# Physical Activity Assessed by Wrist and Thigh Worn Accelerometry and Associations with Cardiometabolic Health

**DOI:** 10.3390/s23177353

**Published:** 2023-08-23

**Authors:** Benjamin D. Maylor, Charlotte L. Edwardson, Alexandra M. Clarke-Cornwell, Melanie J. Davies, Nathan P. Dawkins, David W. Dunstan, Kamlesh Khunti, Tom Yates, Alex V. Rowlands

**Affiliations:** 1Diabetes Research Centre, Population Health Sciences, College of Life Sciences, University of Leicester, Leicester LE1 7RH, UK; bm259@leicester.ac.uk (B.D.M.); melanie.davies@uhl-tr.nhs.uk (M.J.D.); n.dawkins@leedstrinity.ac.uk (N.P.D.); kk22@leicester.ac.uk (K.K.); ty20@leicester.ac.uk (T.Y.); alex.rowlands@leicester.ac.uk (A.V.R.); 2Assessment of Movement Behaviours Group (AMBer), Leicester Lifestyle and Health Research Group, Diabetes Research Centre, University of Leicester, Leicester LE1 7RH, UK; 3Leicester Diabetes Centre, University Hospitals of Leicester NHS Trust, Leicester LE5 4PW, UK; 4School of Health and Society, University of Salford, Greater Manchester M5 4WT, UK; a.m.clarke-cornwell@salford.ac.uk; 5School of Sport and Wellbeing, Leeds Trinity University, Leeds LS18 5HD, UK; 6Physical Activity Laboratory, Baker Heart and Diabetes Institute, Melbourne, VIC 3004, Australia; david.dunstan@baker.edu.au; 7Institute for Physical Activity and Nutrition, Faculty of Health, Deakin University, Geelong, VIC 3220, Australia; 8NIHR Applied Research Collaboration East Midlands, Diabetes Research Centre, University of Leicester, Leicester LE1 7RH, UK

**Keywords:** accelerometry, measurement, GGIR, Axivity, activPAL

## Abstract

Physical activity is increasingly being captured by accelerometers worn on different body locations. The aim of this study was to examine the associations between physical activity volume (average acceleration), intensity (intensity gradient) and cardiometabolic health when assessed by a thigh-worn and wrist-worn accelerometer. A sample of 659 office workers wore an Axivity AX3 on the non-dominant wrist and an activPAL3 micro on the right thigh concurrently for 24 h a day for 8 days. An average acceleration (proxy for physical activity volume) and intensity gradient (intensity distribution) were calculated from both devices using the open-source raw accelerometer processing software GGIR. Clustered cardiometabolic risk (CMR) was calculated using markers of cardiometabolic health, including waist circumference, triglycerides, HDL-cholesterol, mean arterial pressure and fasting glucose. Linear regression analysis assessed the associations between physical activity volume and intensity gradient with cardiometabolic health. Physical activity volume derived from the thigh-worn activPAL and the wrist-worn Axivity were beneficially associated with CMR and the majority of individual health markers, but associations only remained significant after adjusting for physical activity intensity in the thigh-worn activPAL. Physical activity intensity was associated with CMR score and individual health markers when derived from the wrist-worn Axivity, and these associations were independent of volume. Associations between cardiometabolic health and physical activity volume were similarly captured by the thigh-worn activPAL and the wrist-worn Axivity. However, only the wrist-worn Axivity captured aspects of the intensity distribution associated with cardiometabolic health. This may relate to the reduced range of accelerations detected by the thigh-worn activPAL.

## 1. Introduction

Accelerometer-based devices are increasingly being used in research to assess physical behaviour [1]. Traditionally, accelerometers are worn on the hip; however, other wear locations, such as the wrist and thigh, are being increasingly used for large-scale surveillance [1,2]; wrist-wear is associated with higher participant compliance [3], whereas thigh-wear enables more accurate determination of posture and stepping [4,5,6]. Overall, there appears to be a consistent pattern of association between accelerometer-assessed physical activity and health, irrespective of wear location or brand. For example, low levels of physical activity have been associated with all-cause mortality when physical activity is derived from accelerometers worn on the hip [7], wrist [8,9], or thigh [10]. Furthermore, the volume of physical activity, regardless of intensity, has been associated with markers of cardiometabolic health in adults, irrespective of whether the accelerometer is worn on the hip [11,12], wrist [13], or thigh [14].

Accelerometer outputs differ by wear site; thus, intensity and behavioural outcomes are generated using wear-site specific processes (e.g., cutpoints for wrist-worn and posture for thigh-worn accelerometers). Despite wear-specific processes, the magnitude of time spent in behaviours often differs by wear site [15]. Previous studies have reported similar associations of cardiometabolic health for sedentary time derived from thigh- and hip-worn accelerometers [16] and adiposity for physical activity derived from hip- and wrist-worn accelerometers [17]. It is less clear whether this would be the case for directly measured acceleration metrics that facilitate the assessment of the relative contribution of volume and intensity of activity for health.

Physical activity (PA) volume and intensity metrics can be derived directly from accelerometers and can be used to examine associations with health. Average acceleration can be used as a proxy to describe the volume of PA over the 24 h day and the intensity gradient can be used to describe the intensity distribution of PA over the 24 h day [18]. Using this approach, Dawkins et al. [19] reported that both PA volume and intensity were associated with lower cardiometabolic risk in healthy adults, but only for PA volume in those with chronic disease.

These approaches to examine the interrelationship between the volume and intensity of physical activity with health have largely been carried out in studies with wrist-worn accelerometer data. As highlighted previously, the activPAL accelerometer and other brands worn on the thigh are increasingly being used to capture physical activity and sedentary behaviour [20]. Furthermore, accelerometer outputs differ by wear site [21,22], and previous research has demonstrated that the magnitude and distribution of acceleration differs between the activPAL and other common accelerometer brands (Axivity, GENEActiv and ActiGraph) when all are worn concurrently on the thigh [23]. Therefore, the detected associations between the volume and intensity of physical activity and health may differ according to whether data from a thigh-worn activPAL or a wrist-worn Axivity are used to generate the PA metrics.

Therefore, the aim of this study was to examine associations between the intensity and volume of physical activity and cardiometabolic health when physical activity is assessed with the activPAL worn on the thigh and the Axivity worn on the wrist.

## 2. Materials and Methods

### 2.1. Design and Participants

Baseline data from the randomised controlled trial of the SMART Work and Life intervention [24] were used for this cross-sectional analysis. In brief, the SMART Work and Life trial was a three-arm cluster randomised controlled trial assessing the effectiveness and cost-effectiveness of an intervention to reduce sitting time. The trial recruited 756 employees who were predominantly desk-based from six local authorities in the UK. Participants were eligible to take part if they were aged ≥18 years, contracted to work for the local authority at ≥60% full-time equivalent, spent the majority of their day sitting (self-reported), and were able to walk without assistance. Participants were ineligible if they used a height-adjustable workstation, were pregnant or were unable to provide written informed consent. All participants provided written informed consent prior to the baseline measurements. Ethical approval was obtained from the University of Leicester (Ref: 14372) and the University of Salford (Ref: HSR1718-039) prior to the commencement of the study.

### 2.2. Demographics and Anthropometric Measures

Date of birth, sex, ethnicity and postcode were collected by questionnaire. The index of multiple deprivation (IMD) was calculated using the participants’ postcode to determine socio-economic status (SES). Height (Leicester stadiometer), body mass, body fat percentage (Tanita, West Drayton, UK), and waist circumference were measured to the nearest 0.5 cm, 0.1 kg, 0.1%, and 0.5 cm, respectively. Body mass index (BMI) was calculated as body mass (kg)/height (m)^2^.

### 2.3. Metabolic and Cardiovascular Markers

Capillary blood samples were taken following an overnight fast >10 h, and adherence to this was checked verbally by the researcher. Participants who reported not being fasted were excluded from the analysis.

Biochemical outcomes were fasting glucose, triglycerides and lipid profile (HDL-Cholesterol [HDL-C], LDL-Cholesterol [LDL-C] and total cholesterol) calculated using a point-of-care device (Cardiochek Plus, PTS Diagnostics, Whitestown, IN, USA), which has high levels of accuracy when compared with venous sampling [25]. HbA1c was determined using the Quo-Test HbA1c analyser (EKF Diagnostics, Cardiff, UK) from the same capillary sample [26]. Blood pressure was measured three times following a 5 min seated rest. The last two measures were averaged. Mean arterial pressure (MAP) was calculated as:$$MAP≅P_{Dias}+\frac{1}{3}(P_{Sys}-P_{Dias})$$

Our primary outcome was the cardiometabolic risk (CMR) score, which was generated using waist circumference, triglycerides, HDL-C, MAP and fasting glucose [27,28]. After normalisation (log 10), variables were standardised, i.e., 𝑧 = value − mean/𝑆𝐷. HDL-C is protective of cardiometabolic risk, so its z-score was multiplied by −1. Z-scores were summed, and the sum was divided by five to create the CMR score (units of SD). The CMR score is a commonly used measure of cardiometabolic health in studies that have examined physical behaviours, such as sitting, standing and stepping [27,28].

### 2.4. Accelerometer Data Collection and Processing

Participants were requested to wear wrist- (Axivity AX3) and thigh-worn (ActivPAL3 micro) accelerometers simultaneously for 24 h/day for 8 days. Throughout the monitoring period, participants completed a diary that recorded when they got into bed, went to sleep, woke up, got out of bed, and any time when the devices were removed for ≥10 min. Participants also reported whether the day was a workday or not and whether it was a typical day for them.

The Axivity AX3 (Axivity, Newcastle, UK) was initialised to record tri-axial acceleration data at a frequency of 100 Hz with a dynamic range of ±8 g and was worn on the non-dominant wrist using the manufacturer’s wrist strap. The device was initialised, and subsequent data were downloaded in .cwa format using OmGui open-source software (OmGui Version 1.0.0.30, Open Movement, Newcastle, UK). The activPAL3 Micro (PAL Technologies Ltd., Glasgow, UK) was initialised to record tri-axial acceleration data at a frequency of 20 Hz, with a dynamic range of ±2 g. The device was waterproofed with a nitrile sleeve and Hypafix transparent dressing. The device was attached to the midline anterior aspect of the right thigh using an additional piece of Hypafix. Participants were only requested to remove the device if they had a bath or went swimming to avoid damage or loss of the device. The activPAL data were downloaded as a .csv file using PALAnalysis version 8.11 (PAL Technologies Ltd., Glasgow, UK).

On return, data for both devices were visually checked for wear compliance. If there were ≤4 days of wear, participants were contacted and requested to repeat the monitoring period. Downloaded data from both devices were processed separately using open-source R package GGIR version 2.4-0 [29], using default package arguments for signal processing; auto-calibration using local gravity as a reference [30]; calculation of Euclidean Norm minus 1 g (ENMO) from the tri-axial acceleration data averaged over 5 s epochs and expressed in milli-gravitational units (mg); detection of non-wear; and detection of abnormally sustained high values [29]. Days were excluded if there was <16 h data or a calibration error >10 mg. Participants were included in this analysis if they had ≥4 valid days of data from each accelerometer from the same week (participants who re-wore the activPAL to provide enough primary outcome data for the trial were excluded due to the time lag between data from each device) [31].

Average acceleration and intensity gradients were calculated for data from both devices. Average acceleration (hereinafter referred to as volume) represents the average acceleration values across all epochs per 24 h day. Intensity gradient (hereinafter referred to as intensity) describes the distribution of the intensity of accelerations across the 24 h day. The intensity gradient was calculated by accumulating frequencies of acceleration in 25 mg bins, and the regression slope was calculated for the log-transformed variables. Higher values (less negative) indicate a larger proportion of activity spent at a high intensity [32,33]. Additionally, the lowest acceleration values for the most active X (MX) minutes per day were generated for descriptive and interpretative purposes [32]. The thresholds for interpreting moderate-to-vigorous physical activity were 273 mg for the activPAL [34] and 250 mg for interpreting moderate-to-vigorous physical activity intensity indicative of brisk walking for the Axivity [35]. Values were averaged over all valid days.

### 2.5. Statistical Analysis

All statistical analyses were performed in Stata 16 (StataCorp LP, College Station, TX, USA). Continuous parametric participant characteristic data were calculated as mean ± standard deviation (SD), non-parametric data as median (IQR), and categorical data as percentages. Paired *t* tests were used to analyse the differences between summary physical activity metrics from the two accelerometers. Linear regressions were used to assess the correlations between volume and intensity within each device separately. Multiple linear regression was used to assess associations between the volume and intensity derived from the thigh and wrist accelerometers and the cardiometabolic risk score. We undertook secondary analyses on the markers underlying the risk score (CMR, BMI, body fat %, waist circumference, HbA1c, total cholesterol, LDL- and HDL-C, triglycerides, systolic and diastolic blood pressure, and MAP) to show where associations were strongest. Wear site-specific standardised scores were generated for the volume and intensity and the regression coefficients reported per standard deviation (SD) for ease of comparison. Model 1 assessed volume and intensity separately, adjusting for age, sex (male/female), ethnicity (White European/other), smoking status (never/previous/current), lipid-lowering and beta-blocker medication (yes/no), history of type 2 diabetes (yes/no), and deprivation score. Model 2 assessed the independent contributions of volume and intensity by including them both in the same model. Regression coefficients are reported with 95% confidence intervals. Two tailed *p* ≤ 0.05 were considered statistically significant.

Forest tree plots were generated to display regression coefficients from both models using GraphPad Prism version 7.04 (GraphPad Software, San Diego, CA, USA). Radar plots were generated post-hoc to assist with data interpretation using the open-source code RadarPlotGenerator (available at: www.github.com/Maylor8/RadarPlotGenerator (accessed on 13 February 2023)). This utilises package ggplot2 in R and has been described previously [32].

## 3. Results

Of 659 participants, 610 (80.6%) (mean age (±standard deviation): 44.6 ± 10.4 years; 72.9% female; mean BMI: 26.4 ± 6.1 kg/m^2^) were included in the final analyses. Appendix A describes the flow of participants through the study. The characteristics of the included participants are provided in Table 1. There were no differences in any of the characteristics between those who provided valid data for these analyses and those who did not.

Table 2 shows the correlation and difference between the activPAL and Axivity derived summary variables. Correlations between volume and intensity were significant and moderate when assessed by the thigh-worn activPAL (r = 0.543, *p* < 0.001) and wrist-worn Axivity devices (r = 0.570, *p* < 0.001), indicating that the volume and intensity provided overlapping and complementary information. More valid days were obtained from the wrist-worn Axivity compared with the thigh-worn activPAL (mean (±SE) difference −0.23 ± 0.04 days, *p* < 0.001). The volume (mean (±SE) difference −4.81 ± 0.22 mg, *p* < 0.001) was higher with the wrist-worn Axivity compared with the thigh-worn activPAL. Conversely, the intensity was higher when derived from the thigh-worn activPAL (mean (±SE) difference 0.49 ± 0.06 units, *p* < 0.001) compared with the wrist-worn Axivity.

Figure 1 displays the associations between physical activity volume and intensity assessed by thigh-worn activPAL and wrist-worn Axivity, and cardiometabolic health risk (bottom of plots) and individual markers. After adjustment for basic confounders (model 1), both activPAL- and Axivity-derived volume were significantly and beneficially associated with CMR score, whereas only intensity derived from the wrist-worn Axivity was significantly and beneficially associated with CMR score. Following additional adjustment for intensity (model 2), volume remained significantly associated with CMR when derived from the thigh-worn activPAL, but not the wrist-worn Axivity. Conversely, after adjustment for volume, the intensity remained significantly associated with wrist-worn Axivity. Regression coefficients (unadjusted and adjusted) are detailed in Appendix A.

For individual cardiometabolic health markers following adjustment for basic confounders (model 1), volume when derived from either accelerometer was significantly associated with HbA1c, HDL cholesterol, BMI, waist circumference, body fat %, diastolic blood pressure (DBP) and MAP. Additional associations were seen for volume derived from the thigh-worn activPAL for triglycerides and systolic blood pressure. The magnitudes of association were similar for the thigh-worn activPAL and wrist-worn Axivity across all variables. For intensity, the only significant association observed for both thigh-worn activPAL- and wrist-worn Axivity was for HDL cholesterol, whereas wrist-worn Axivity-derived intensity was also associated with all other markers except for glucose, LDL and total cholesterol. ActivPAL-derived intensity was not associated with any other markers of health.

Following adjustment for intensity (model 2, Figure 2), all associations for volume observed in model 1 remained significant, though associations were weaker than model 1 when derived from the Axivity and stronger when derived from the activPAL. Following adjustment for volume, all associations for intensity observed in model 1 remained significant but weaker, with the exception of the thigh-worn activPAL and HDL cholesterol.

### MX Results

To aid in the interpretation of the differences in accelerations between the thigh-worn activPAL and wrist-worn Axivity, the MX values were plotted on a radar plot (Figure 3). This shows that the distribution of acceleration differed by device/wear-site. Higher acceleration values were generated by the wrist-worn Axivity during the most active very short periods (1–2 min) of the day and over the most active long duration periods (2–12 h) of the day. However, higher acceleration values were observed during the most active 10–60 min of the day for the thigh-worn activPAL. These contrasts in the distribution of acceleration by device/wear-site are indicated by the crossing of the red and blue lines towards the bottom left and top left of the plot. Appendix A displays the MX values for each device.

## 4. Discussion

This study is the first to model the associations of two cutpoint-independent physical activity volume and intensity metrics with cardiometabolic risk scores and individual markers of health when derived from thigh and wrist accelerometers. The key findings from this analysis are that physical activity volume, derived from the thigh-worn activPAL or wrist-worn Axivity, was associated with various cardiometabolic risk markers (model 1), but associations were independent of intensity only for the thigh-worn activPAL (model 2). In contrast, physical activity intensity was only associated with cardiometabolic risk markers when derived from the wrist-worn Axivity and associations were independent of volume. This suggests that the intensity distribution derived from the thigh-worn activPAL did not capture the aspects of intensity associated with cardiometabolic health. This may relate to the reduced range of accelerations detected by the thigh-worn activPAL as recently shown in children by Buchan et al. [15]. The strength of associations between physical activity volume and markers of health were similar across the wrist and thigh worn accelerometers but were three times stronger for physical activity intensity and markers of health when derived from the wrist accelerometer.

Comparisons with previous research are challenging, as there is limited evidence investigating the associations between the newer physical activity volume and intensity metrics used in the present study and cardiometabolic risk markers. One recent study by Backes et al. [36] examined associations between intensity gradient and average acceleration, assessed by a wrist-worn ActiGraph accelerometer, and insulin sensitivity and glycated haemoglobin and found that changes to both variables were associated with improved insulin sensitivity, but not HbA1c after adjusting for similar confounding variables as the present study. Likewise, the present study observed that intensity gradient was not associated with HbA1c, although average acceleration was without additional adjustment for intensity gradient. When considering physical activity volume and intensity more broadly, similar to the present study, physical activity volume and intensity appear to be associated with measures of adiposity, such as body fat percentage [37], BMI and waist circumference [38], regardless of the device and metrics used to produce volume and intensity. However, for other metrics of cardiometabolic health (e.g., fasting glucose, blood pressure, LDL cholesterol, triglycerides), no associations between physical activity volume, when described as steps per day, and intensity have been reported [38,39], with the exception of daily MVPA time and HDL cholesterol [39]. In the current study, we also observed that intensity (assessed by either accelerometer) was associated with HDL cholesterol but not after adjusting for volume. Similar to previous research, we also found no associations between intensity and other cardiometabolic markers, with the exceptions being triglycerides and the overall cardiometabolic risk score. Our results for physical activity volume were contrasting across our two accelerometers, with activPAL-assessed volume being associated with an overall cardiometabolic risk score, triglycerides, HDL, total cholesterol and blood pressure, after adjusting for confounders including physical activity intensity, but Axivity-assessed volume was not associated with any cardiometabolic markers, apart from the adiposity ones highlighted previously.

As well as wear-site, there are two inherent disparities that likely contributed to the differences between the two accelerometers used in the present study. First, the Axivity recorded at a sampling rate of 100 Hz and with a dynamic range of ±8 g compared with the 20 Hz sampling rate and ±2 g dynamic range of the activPAL3 micro model, making it likely that more intense accelerations were blunted with the activPAL. Indeed, Small et al. [40] recently reported a <14% reduction in acceleration values when wrist accelerometer data were recorded at 25 Hz compared with 100 Hz. When both the Axivity and activPAL were worn concurrently on the thigh, using the same recording specifications as the present study, Edwardson et al. [23] observed lower (23%) values for activPAL assessed average acceleration compared with the Axivity. However, in contrast to the current study, the activPAL assessed intensity gradient was also lower, suggesting that the higher intensity gradient observed in the current study is due to the thigh wear-site, not the accelerometer. Therefore, it is likely that the intensity distribution measured at the thigh is lower than that measured at the wrist. This is highlighted in the radar plot illustrating the MX distribution (Figure 3), which shows that the accelerations were higher in the wrist-worn Axivity than the thigh-worn activPAL for the most active 2 to 12 h, which represents the majority of waking hours. Furthermore, the intensity of the most active 1–5 min was also higher at the wrist, with the maximum intensity recorded 24.6% higher at the wrist. Conversely, the M5 to M60 accelerations were higher in the thigh-worn activPAL compared with the wrist-worn Axivity. However, the intensity gradient calculated from activPAL data was not associated with CMR. We observed a higher average acceleration across the 24 h day from the wrist accelerometer. This could, in part, be due to the capture of wrist movements, meaning that the wrist-worn device had greater sensitivity to upper body movements, which may be an important contributor to cardiometabolic risk markers such as body composition, lipid and glucose metabolism [41]. Additionally, the capture of higher accelerations detected by the wrist monitor (for example, during M1–M5) also contributed to a higher overall volume.

The sample used in the present study were office workers, so a high proportion of the waking day is likely to include participants who sat at their desk and thereby used their hands to operate their computer while the lower half of the body remained relatively still. Conversely, accelerations captured on the thigh are likely more indicative of purposeful movements, such as walking, a behaviour associated with beneficial health outcomes [42]. Despite this, acceleration values for walking are higher on the wrist than the hip [21] and we anticipated that these differences would have been even larger between the wrist and thigh accelerometer wear locations. Separate analyses of the thigh-worn activPAL data reported participants spending a mean of 109 min/day stepping [24]. However, we observed higher acceleration values in the thigh for M5 to M60 durations (Figure 3) when we expected the wrist to capture higher accelerations. It is possible that confounding factors affected this, such as participants walking with their hand in their pocket or holding their phone/bag and not swinging their wrist in a typical manner, as instructed during laboratory testing. In the present study, higher accelerations were observed for M1 and M2 s with the wrist-worn Axivity, which suggests the Axivity captured a wider intensity distribution than the thigh-worn activPAL and therefore associations of cardiometabolic health markers and the intensity gradient derived from a thigh-worn activPAL may lack utility.

### Strengths and Limitations

The main strengths of this study were the comparison of data-driven metrics derived from two concurrently worn research-grade accelerometers that are widely used in the field of sedentary behaviour and physical activity research. However, there are some limitations to our analysis that should be noted. First, we did not down-sample the Axivity data to 20 Hz in order to match the activPAL, which might have assisted in determining how much difference was due to the recording specifications or wear-site of the different monitors. However, we used the default recording settings for both monitors, as is commonly adopted by researchers. This makes the findings more generalisable and comparable to the majority of existing data collected using these monitors and wear sites. A second limitation was that we conducted multiple comparisons across the individual markers of health, which increases the risk of a type 1 error. However, our primary outcome was the clustered cardiometabolic risk score, with the additional comparisons used as explanatory contributions towards the clustered statistics. A third limitation was that although the sample was relatively large, the markers of health suggested the sample was relatively healthy. This may have weakened the strength of the associations in our analyses. The use of these metrics in samples at a higher risk of cardiometabolic disease or with a sample covering different occupations should be investigated. Additionally, the study assessed data cross-sectionally and needs further confirmation in longitudinal studies.

## 5. Conclusions

Despite differences in magnitude, physical activity volume was similarly associated with cardiometabolic health markers for both devices, although the association was only independent of intensity when derived from the thigh-worn activPAL. Conversely, the intensity distribution was only associated with cardiometabolic health when derived from the wrist-worn Axivity and associations were independent of volume. The lack of an association between the intensity measured at the thigh and cardiometabolic health may relate to a reduced range of accelerations detected by the thigh-worn activPAL. This suggests that the intensity gradient measured at the thigh may not provide meaningful information in relation to cardiometabolic health.

## Figures and Tables

**Figure 1 sensors-23-07353-f001:**
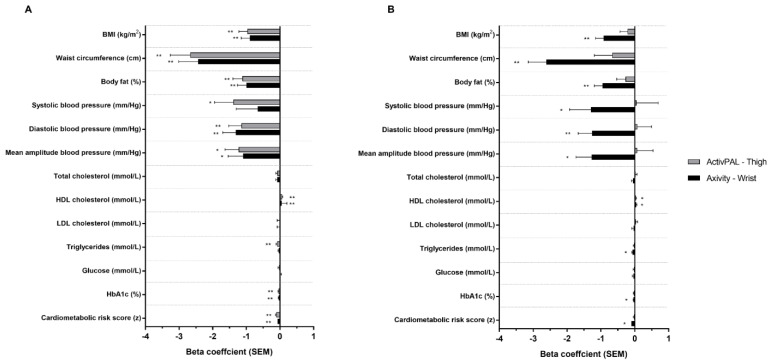
Associations between ActivPAL and Axivity average acceleration (**A**) and intensity gradient (**B**) with cardiometabolic risk score and individual health markers. Adjusted for age, sex, ethnicity, smoking status, medical history of type 2 diabetes, lipid lowering medication or beta blockers and deprivation score (model 1). Beta coefficients are standardised. * *p* < 0.05; ** *p* < 0.001.

**Figure 2 sensors-23-07353-f002:**
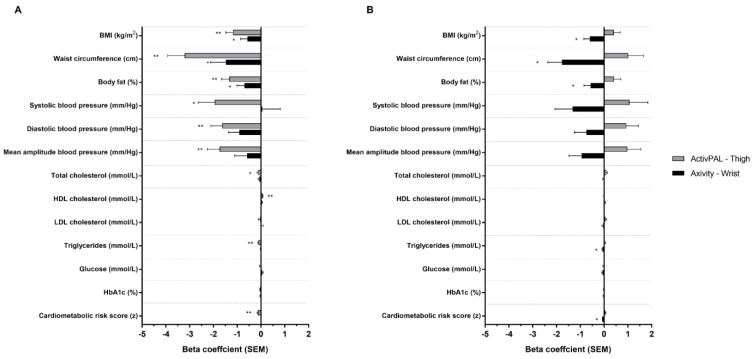
Associations between ActivPAL and Axivity average acceleration (**A**) and intensity gradient (**B**) with cardiometabolic risk score and individual health markers. Adjusted for intensity gradient (**A**) or average acceleration (**B**), age, sex, ethnicity, smoking status, medical history of type 2 diabetes, lipid lowering medication or beta blockers and deprivation score (model 2). Beta coefficients are standardised. * *p* < 0.05; ** *p* < 0.001.

**Figure 3 sensors-23-07353-f003:**
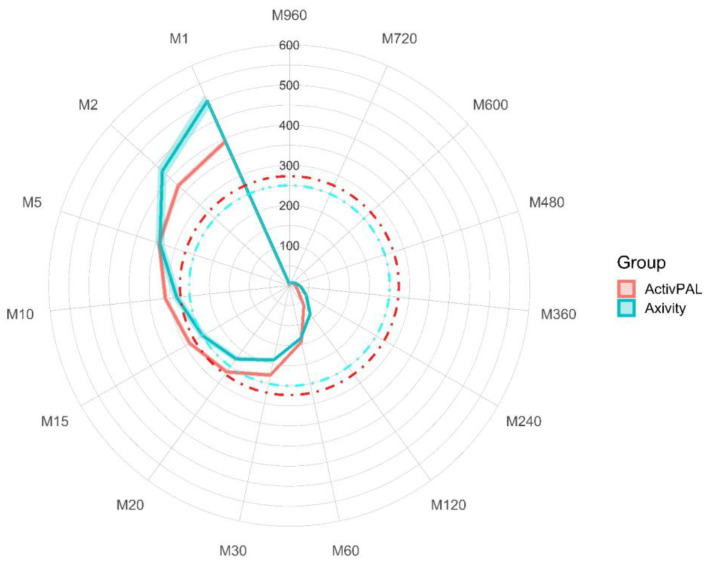
MX values for both devices. Mean MX values are expressed as milli-gravitational units. Shaded error represents 95% confidence intervals. The red dashed line is indicative of a moderate-to-vigorous activity threshold for the ActivPAL, and the blue dashed line is indicative of a brisk walking pace for the Axivity. MX, most active X minutes from any point in the 24 h day.

**Table 1 sensors-23-07353-t001:** Participant characteristics.

Characteristic	Mean (SD)
Age (years)	44.6 (10.4)
Sex, *n* women	445 (72.9%)
Ethnicity, *n* White European	432 (70.8%)
Cardiometabolic risk score (z)	−0.01 (0.66)
Body Mass Index (kg/m^2^)	26.5 (5.9)
Fasting Glucose (mmol/L)	5.49 (0.96)
HbA1c (%)	5.25 (0.49)
Total cholesterol (mmol/L)	4.70 (1.07)
HDL-C (mmol/L)	1.43 (0.40)
LDL-C (mmol/L)	2.59 (1.10)
Triglycerides (mmol/L)	1.22 (0.64)
Systolic blood pressure (mmHg)	117.9 (15.9)
Diastolic blood pressure (mmHg)	79.1 (10.3)
Mean Amplitude of blood pressure (mmHg)	92.1 (11.6)

**Table 2 sensors-23-07353-t002:** Accelerometer summary statistics.

Variable	Axivity	ActivPAL	Mean Axivity-Activpal Delta	*p*
Number of valid days	7.8 (0.5)	7.6 (0.9)	0.2 (1.1)	<0.001
Average acceleration (mg)	27.40 (7.11)	22.52 (6.58)	4.81 (5.71)	<0.001
Intensity gradient	−2.55 (0.21)	−2.05 (0.22)	−0.49 (0.26)	<0.001
Intensity gradient R^2^	0.89 (0.04)	0.81 (0.07)	−0.08 (0.06)	<0.001
Average acceleration—Intensity gradient Correlation	0.570	0.543	-	-

## Data Availability

The data that support the findings of this study are not openly available due to them containing information that could compromise research participant privacy/consent. Requests for participant-level quantitative data and statistical codes should be made to the corresponding author. Data requests will be put forward to members of the original trial management team who will release data on a case-by-case basis.

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
