# Peer review of "Physical Activity Assessed by Wrist and Thigh Worn Accelerometry and Associations with Cardiometabolic Health"

_sensors, 2023, doi:10.3390/s23177353_

Round 1
Reviewer 1 Report
Overall, this is a well written manuscript that aimed to examine associations between physical activity volume, intensity and cardiometabolic health when assessed by thigh-worn and wrist-worn accelerometer. The article does provide a novel insight into the comparison of wrist vs thigh worn accelerometer on CMR. However, there are a few considerations to be given to statistical analysis with respect to Type 1 errors and classification of the population grouping
Specific points:
Table 1: insert p values to signify differences in data collection between monitors. Within accelerometer comparison I would suggest representing the delta values rather than group means as this does not allow us to see the individual/paired characteristics of the data.
Line 212: remove less negative
Please indicate/describe how the author has mitigated against type 1 errors in analysis as there is a large number of associations analysis
Throughout text please clarify is it p<0.05/p<0.001 or is it p≤0.05/p≤0.001
More valid days were obtained from the wrist-worn Axivity compared with the thigh-worn activPAL. Was data collection standardised for a set number of days? Was there greater variation seen within the wrist worn in comparison to thigh-worn? Can you please include in your methodology how you standardized collection across the time collected and if a varying volume of capture data impacted the variation in the groups.
Discussion: Individuals within the studies presented (on average- table 1) with low CMR and had relative risk factors within clinical range. Therefore, the data presented here would only correspond to individuals of low risk. Previous work from the group (DOI: 10.1249/MSS.0000000000002939) has published differing associations of intensity and volumes dependent on disease status.
Please can the author confirm some points and address in text if necessary: Are the associations reported here independent of disease? Across the 659 participants are all considered healthy individuals- if so, the title needs inclusion of CMR risk in healthy individuals?
The author does mention this very briefly in limitations however this is a major consideration, and the author needs to address the significance this has on the findings and further impact it may have i.e. if individuals are healthy then would the data have the true range to indicate association. Also, what is the importance or rationale for establish device location specific associations of disease risk, in those with very limited risk?
Author Response
Thank you for your review. Please see our responses in the attached document.

Reviewer 2 Report
This paper proposed a method of assess physical activity volume used a thigh-worn and a wrist-worn accelerometer. The topic is interesting, but the reviewer thought the paper is not fit for the MDPI Sensors journal.
1. It's better for the author to put some figures of the experiment, and better describe the experiment protocal.
2. It's better for the author to give a scientific method, the current version seems like an investigation.
3. It's better for the author to give a scientific method of the paper, the current version seems like a statistical study. 4. What's the two kinds of sensor modules used for? What kinds of parameters of the authors collected? For example, 3-axis of accelerometer, 3-axis of gyro etc, the author didn't mention about it. 5. The authors need to provide a flowchars of their method. 6. How you process the data you collected from the 2 kinds of sensors, the author didn't mention about it. 7. It's better for the author to put some figures of the experiment, and better describe the experiment protocal. 8. Some abbr words you first mentioned in the paper should give the full name of the words. For example, in the abstract "GGIR".Author Response
Thank you for your review. Please see our responses in the attached document.

Reviewer 3 Report
Thank you for your submission. The manuscript was well-presented and easy to follow. Notably, the Materials and Methods section contained a lot of detail, making the steps involved very clear. The discussion also laid out each pertinent issue effectively for the reader.
Ideally, the two monitors would have been more “evenly” matched in the recording frequency and thresholds. However, both accelerometers used are of adequate quality. It was stated that the physical activity volume was higher with the wrist-worn monitor, but the actual difference was not given. I think it would be interesting to provide the actual average acceleration and intensity gradient values for each monitor, especially considering the potential role the difference in collection frequency and range may have played.
There is a minor but repeated issue with word choice: the wording, “…associations were independent of intensity only for the thigh-worn activPAL. In contrast, physical activity intensity was only associated with cardiometabolic risk markers when derived from the wrist-worn Axivity…” seemed redundant. The restatement of this result appears in the abstract, discussion (lines 274-277), and conclusion. With only two accelerometers, the use of the word “only” is not necessary. And if I am not misreading them, the two sentences effectively state the same thing, that physical activity intensity is related to the wrist-worn, but not the thigh-worn accelerometer.
Though outside of the parameters of the study, the willingness of individuals to wear a wrist- vs thigh-worn accelerometer is also something to consider regarding the application of the study. If one is better, but people will not use it, then it is no longer better. Inclusion of a questionnaire about comfort and/or willingness to use these devices could prove to be a valuable addition.
Author Response

(The authors gave the same response as above.)

Round 2
Reviewer 2 Report
The authors has reply all the comments that the reviewer has proposed.
It could be accepted in this version.